# Childhood Adversities and the ATTACH^TM^ Program’s Influence on Immune Cell Gene Expression

**DOI:** 10.3390/ijerph21060776

**Published:** 2024-06-14

**Authors:** Zhiyuan Yu, Steve Cole, Kharah Ross, Martha Hart, Lubna Anis, Nicole Letourneau

**Affiliations:** 1School of Nursing, Johns Hopkins University, 525 N Wolfe St., Baltimore, MD 21205, USA; zyu46@jhu.edu; 2School of Medicine, University of California-Los Angeles, Le Conte Ave, Los Angeles, CA 10833, USA; steve.cole@ucla.edu; 3Department of Psychology, Athabasca University, 1 University Dr., Athabasca, AB T9S 3A3, Canada; kharahr@athabascau.ca; 4Alberta Children’s Hospital Research Institute, University of Calgary, 2500 University Drive NW, Calgary, AB T2N 1N4, Canada; mhart@ucalgary.ca (M.H.); lanis@ucalgary.ca (L.A.); 5Faculty of Nursing & Cumming School of Medicine, University of Calgary, 2500 University Drive NW, Calgary, AB T2N 1N4, Canada

**Keywords:** adverse childhood experiences, parenting intervention, ATTACH™, immune cell gene expression, CTRA

## Abstract

Objective: To determine whether maternal Adverse Childhood Experiences (ACEs) are (a) associated with increased inflammatory gene expression in mother–child dyads and (b) whether a parenting intervention (ATTACH™) moderates the association between maternal ACEs and mother and/or child inflammatory gene expression. Methods: Twenty mother–child dyads, recruited from a domestic violence shelter in Calgary, AB, Canada, were randomized into an ATTACH™ parenting intervention group (*n* = 9) or a wait-list control group (*n* = 11). Maternal ACEs were assessed. The mothers and children each provided one non-fasting blood sample after the intervention group completed the ATTACH™ program, which was assayed to quantify the Conserved Transcriptional Response to Adversity (CTRA) score, indicating inflammatory gene expression profile. Mixed-effect linear models were used, separately in mothers and children, to examine the associations between CTRA score, maternal ACEs, and the ACEs-by-intervention group interaction term. The covariates were age, sex, ethnicity, and maternal medication use. Results: Higher maternal ACEs were associated with higher child CTRA scores (*b* = 0.123 ± SE 0.044, *p* = 0.005), indicating an increased pro-inflammatory gene expression profile. The ATTACH™ parenting intervention moderated this association between maternal ACEs and child CTRA scores (*b* = 0.328 ± SE 0.133, *p* = 0.014). In mothers, the ACEs-by-intervention interaction terms were insignificant (*p* = 0.305). Conclusions: Maternal ACEs could exert an intergenerational impact on child inflammatory activity, and this association could be moderated by participating in the ATTACH™ parenting intervention.

## 1. Introduction

Adverse Childhood Experiences (ACEs) are potentially traumatic events that occur before 18 years of age, including child abuse, neglect, and exposure to stressful living environments (e.g., domestic violence, parental mental illness, and substance use). ACEs are linked to subsequent risk for a wide range of poor health outcomes in adulthood, particularly cancer, and neuropsychiatric, cardiometabolic, and sleep disorders [1,2,3,4,5,6,7,8,9]. Growing studies suggest that immune system dysregulation, including chronic inflammation, may be one of the key biological mechanisms through which ACEs affect illness or poor health in later life. ACEs could lead to prolonged activation of the biological stress responses, such as sympathetic nervous system (SNS), increasing inflammatory burden. This, in turn, may “get under the skin” and elevate the risk of poor health and morbidity well into adulthood [5,10,11,12]. Given the mediating role of immune regulation, identifying factors that buffer the effects of ACEs on inflammatory biology may help mitigate the risks of illness among children and families exposed to high levels of adversity.

One indicator of inflammatory activity is the Conserved Transcriptional Response to Adversity (CTRA), a conserved response to social adversities, including loneliness, poverty, bereavement, chronic stress, and ACE exposures [13,14,15]. The CTRA is characterized as a pattern of immune gene expression defined by up-regulated pro-inflammatory gene expression and the down-regulated expression of genes involved in Type I interferon response [15]. The CTRA is also associated with poor health outcomes, including symptoms of chronic fatigue and depression [16,17]. There is evidence that child ACE exposure is associated with child CTRA scores. In a study of 37 healthy children (aged 5 to 11 years), higher child ACEs exposure was associated with higher CTRA scores, specifically elevated pro-inflammatory and Type-I interferon gene expression, consistent with a skew towards pro-inflammatory activity [18]. Although higher adult ACE exposures are associated with inflammatory outcomes, no studies were found that examined associations between mothers’ ACEs and their CTRA scores, or mothers’ ACEs and child CTRA scores.

Sensitive, responsive parenting may help buffer the effects of ACEs on children [19,20], but parents’ own history of ACEs can challenge their ability to practice these optimal parenting behaviors. Higher maternal ACEs are associated with poorer child outcomes [21], particularly child mental health problems, and child externalizing and internalizing behavioral difficulties [22], likely through negative impacts on parenting. Mothers’ ACEs can disrupt their emotional regulation, parenting behavior, parent–child attachment, and capacity to provide sensitive, responsive parenting, despite their desire to provide the best care to their children [23,24,25,26]. Less sensitive (negative or unpredictable) parenting puts children at risk for immune dysregulation [27,28,29]. Thus, mothers’ ACEs may impose immunological risks on their children through less sensitive and responsive parenting. 

If maternal ACEs affect parenting behaviors and skills, with adverse implications for children, then parenting interventions could theoretically mitigate the effects of maternal ACEs on child outcomes. The Attachment and Child Health (ATTACH™) parenting program was developed to support sensitive, responsive parenting for families affected by adversity, such as maternal depression, domestic violence, or low income. ATTACH™ focuses on improving parents’ reflective function, that is, the parent’s ability to think about and identify mental states, thoughts, feelings, and intentions in themselves and in their children [30,31]. To date, published studies have demonstrated positive impacts of ATTACH™ on parent–child interaction quality, attachment security, parental reflective function, executive function, children’s development (particularly communication, problem-solving, personal–social skills, and fine motor skills), children’s sleep, children’s behavioral problems (e.g., anxiety/depression, attention, aggression) and inflammatory gene expression in both parents and children’s immune cell gene expression [32,33]. Further, it is documented that families with higher risks tend to benefit more from parenting interventions [34], and that the benefits may be mediated by improved parenting quality [35]. Whether a parenting intervention, like ATTACH™, could moderate associations between maternal ACEs and mother–child inflammatory outcomes, like the CTRA, has not been tested. 

The purpose of this study was to address the gaps identified above, by examining whether maternal ACEs are associated with mother and/or child CTRA scores, and whether a parenting intervention (ATTACH™) moderates the association between maternal ACEs and mother and/or child CTRA scores. We hypothesized that: (1) higher maternal ACEs would be associated with higher CTRA scores in mothers and children, and (2) completing an established and effective parenting intervention (ATTACH™) would buffer or attenuate associations between maternal ACEs and mother and child CTRA scores. 

## 2. Methods

### 2.1. Participants

Twenty mother–child dyads were recruited from a domestic violence shelter in Calgary, AB, Canada. Mothers were eligible for the study if they were able to read and write in English, were the primary caregiver of a child less than six years of age, and were not planning to relocate in the next three months. Over a third of the mothers were Indigenous (35%), followed by those with White/Caucasian (30%), Asian (15%), African American/Black (10%), Hispanic or Latin or Middle East origins (10%). The average ages of mothers and children were 31.5 years (SD = 4.9) and 40.4 months (SD = 32.8), respectively. Over half (55%) of the children were female. Detailed sample characteristics are described elsewhere [33].

### 2.2. Procedure

Families were randomized to either an intervention group (*n* = 9) or a wait-list control group (*n* = 11). The intervention group was provided with the ATTACH^TM^ intervention immediately. In contrast, the wait-list control group was provided with the ATTACH™ intervention after the intervention group had undergone the intervention and post-intervention assessment. Mothers received care as usual, including social support through the shelter staff and additional programming, throughout the study. Mothers and children provided one-time dried blood spots (DBS) after the intervention group completed the ATTACH^TM^ intervention, and before the wait-list control families began the ATTACH™ intervention. At the time of blood sampling, none of the mothers reported the presence of any acute infection at blood sampling. Additionally, there was no reported use of medication among the children. This study received institutional review board approval (ethics ID: REB14-0368) from the University of Calgary.

### 2.3. The ATTACH^TM^ Program

The ATTACH™ program is described in more detail elsewhere [32]. Briefly, the program is composed of a brief 10-week psychoeducational parenting intervention with dyadic (mother and child) and triadic (mother, child, and co-parenting support person) components to foster parental reflective function through practice. The program is broadly targeted towards families affected by adversity, without being tailored to a specific client population. It is designed to be minimally burdensome, provided in addition to or as a complement to existing agency programming, and deliverable by community agency staff. Weekly sessions include a focus on learning and practicing reflective functioning skills. The sessions consist of leading by example, asking questions and providing opportunities to reflect on actual mother–child interactions during the session via a videotaped caregiver/child free-play, followed by a hypothetical and real-life situation. After establishing a therapeutic relationship through six one-on-one therapy sessions, parents are encouraged to invite a friend or family member to join the intervention as a co-participant, providing additional parenting support. These “co-parents”—such as grandparents, relatives, friends, or other support person—attend 2–3 sessions, typically sessions 7 and 9, spaced 2 weeks apart. 

### 2.4. Measures

#### 2.4.1. Adverse Childhood Experiences

The Adverse Childhood Experiences (ACEs) scale [2] consists of 10 questions that query the extent to which a participant was exposed to early childhood adversity. Participants are asked to respond by thinking about events prior to their 18th birthday. Example items are “Did you live with anyone who was depressed, mentally ill, or attempted suicide?” and “Did a parent or adult in your home ever hit, beat, kick, or physically hurt you in any way?” Questions are responded to as yes (1) or no (0). ACEs totals are calculated by summing responses across all questions, with scores ranging from 0 to 10 and higher scores indicating greater exposure to childhood adversities. The ACEs scale demonstrated good reliability in this study with the Cronbach’s α = 0.78.

#### 2.4.2. Inflammatory Gene Expression

A non-fasting blood sample was collected from the parent and child onto a DBS card and assayed for genome-wide transcriptional profiles, as described previously [33]. Briefly, genome-wide transcriptional profiling was conducted by RNA sequencing, and data were quantified as gene transcripts per million total transcriptome-mapped sequencing reads. Then, these data were normalized to equate the expressions of 11 human reference genes [36], with a floor set at 1 transcript per million to suppress spurious variance. Subsequently, data were log2-transformed to stabilize variance and screened to exclude transcripts with less than 0.5 log2 units across participants (removing genes that were generally undetectable or showed no appreciable variation in expression levels), and the z-scores were standardized within each gene to facilitate analysis using linear statistical models, as outlined below.

#### 2.4.3. Covariates

Covariates that could be related to immune cell gene expression patterns were also measured using a demographic questionnaire, including age, sex, ethnicity, and maternal medication use (e.g., medication for asthma, anti-depressant, thyroid, and diabetes). 

### 2.5. Data Analysis

Gene expression data were analyzed as previously described [33], using a standard (pre-specified) set of gene transcripts used in previous research to quantify CTRA RNA profile [13,14,15]. The CTRA profile was quantified by the average expression of an a priori specified set of 19 pro-inflammatory gene transcripts (e.g., *IL1B*, *IL6*, *IL8/CXCL8*, *COX2/PTGS2*, *TNF*) and 34 Type I interferon- and antibody-related gene transcripts (e.g., *IFNB*, *IRF7*, *IFI27*, *MX1*, *OAS1*, etc.; all of which were sign-inverted to reflect their inverse contribution to the CTRA profile) [15]. Here, 6 of the pre-specified indicator genes were excluded from for analysis due to minimal variation (SD < 0.5 log2 units), resulting in a total of 47 CTRA indicator genes available for analysis. Gene expression data from parent and child DBS samples were analyzed separately using mixed effect linear models. These models treated the 47 available CTRA indicator genes as a repeated measure and specified a random subject-specific intercept nested within a random family-specific intercept to account for association among residuals across the repeated measures. 

To determine whether maternal ACEs were associated with increased CTRA scores in mothers and children, an initial “marginal effects analysis” (Model 1) quantified the association between maternal ACEs total score and CTRA scores in mothers and children, respectively, while controlling for intervention condition (ATTACH™ vs. Control). To determine whether completing ATTACH™ would buffer or attenuate associations between maternal ACEs and mother and child CTRA scores, Model 2 added an ACEs-by-intervention group interaction term, with a follow-up simple slopes analysis quantifying the association of CTRA scores with maternal ACEs separately for participants in the ATTACH™ intervention vs. Control groups. A secondary set of analyses also controlled for covariates.

## 3. Results

Maternal ACEs total scores did not significantly differ between the intervention and wait-list control groups (Intervention mean = 4.2; Control mean = 5.1; difference, *p* = 0.439). 

### 3.1. Maternal ACEs and CTRA Gene Expression

Mixed effect linear models were used to test the association between maternal ACE scores and the average expression of CTRA indicator gene transcripts (i.e., CTRA scores) in mothers and children, respectively, controlling for any effects of the ATTACH^TM^ intervention. In mothers, the associations between maternal ACEs and CTRA scores were marginally significant (+0.102 log2 mRNA abundance per ACEs Total Score unit ± SE 0.056, *p* = 0.069), such that there was a trend of higher maternal ACEs being associated with higher maternal CTRA scores. In children, higher maternal ACEs was significantly associated with higher child CTRA scores (*b* = 0.119, SE = 0.051, *p* = 0.020). As a sensitivity test, we controlled for additional sources of variability by adding age, ethnicity, medication exposures, and (in the case of the child) child sex, but the patterns of results were unchanged (mothers +0.078 ± 0.060, *p* = 0.195; children +0.123 ± 0.044, *p* = 0.005; Figure 1).

### 3.2. Maternal ACEs and ATTACH^TM^ Intervention Effects on CTRA Gene Expression

To determine whether the ATTACH™ intervention could moderate associations between maternal ACEs and CTRA scores, separate models were run predicting mother or child CTRA scores from maternal ACEs, intervention group, and the ACEs-by-intervention interaction term. In mothers, the ACEs-by-intervention group interaction term was not significant, with or without covariates in the model (unadjusted, +0.011 ± 0.112, *p* = 0.922; covariate-adjusted, +0.134 ± 0.130, *p* = 0.305). 

In children, however, the ACEs-by-intervention group interaction term was significant (unadjusted, +0.207 ± 0.091, *p* = 0.023; covariates-adjusted, +0.328 ± 0.133, *p* = 0.014). As shown in Figure 2, the association between maternal ACEs and child CTRA scores was attenuated for the ATTACH™ intervention group, as compared to the wait-list control group. For participants in the wait-list control group, the association between maternal ACEs and child CTRA was non-significant (*b* = −0.028, *p* = 0.707). However, a significant association between maternal ACEs and child CTRA emerged for the ATTACH intervention group (*b* = 0.300, *p* < 0.001), such that lower maternal ACE scores were associated with lower child CTRA scores, suggesting that the intervention was more effective for children of mothers less affected by early life adversity.

## 4. Discussion

The purpose of this study was to determine whether maternal ACEs were associated with mother or child inflammatory activity, as captured by CTRA scores, and whether a parenting intervention (ATTACH™) could moderate or buffer any association between maternal ACEs and mother or child CTRA scores. Consistent with the hypotheses, higher maternal ACEs were associated with higher child CTRA scores, indicating an increased pro-inflammatory gene expression profile. Also consistent with the hypotheses, the ATTACH™ parenting intervention moderated the association between maternal ACEs and CTRA scores, but not necessarily by buffering the effects of maternal ACEs on CTRA scores. Instead, the results suggest that lower maternal ACEs are associated with better responses to the ATTACH™ intervention, as evidenced specifically by lower child CTRA scores. These findings support the hypothesis that ACE exposures could affect future health outcomes through pro-inflammatory-skewed immune activity and capture the intergenerational transmission of ACE exposure, and suggest that the effectiveness of parenting interventions could vary by parental exposure to early life adversity.

Maternal ACEs were associated with elevations in child CTRA gene expression, indicating an increased pro-inflammatory gene expression profile, independent of the previously documented association between the ATTACH™ intervention and mother and child CTRA scores [33]. These results are consistent with evidence suggesting an intergenerational impact of ACEs, such that parental ACEs could affect child biopsychological and developmental outcomes, such as cardiometabolic (e.g., higher systolic blood pressure), and emotional (e.g., anxiety) and behavioral (e.g., Attention Deficit/Hyperactivity Disorder), health problems [21,22,26,37,38]. To our knowledge, previous studies have not examined associations between parental ACEs and child immune or inflammatory outcomes. Our results suggest that ACEs have the potential to exert intergenerational effects, such that the maternal experience of ACEs could be passed on to affect child immune activity, which is known to affect heath across the lifespan [39]. Moreover, the pro-inflammatory nature of the CTRA suggests one potential explanation for previously observed epidemiological associations between parental ACEs exposure and child physical and mental health outcomes.

Of interest, although there was a trend towards higher maternal ACEs associated with higher maternal CTRA, the association did not achieve statistical significance. This is consistent with another study of 259 sexual minority Black and Latino men, for which no association was found between individual or total ACE exposures and adult CTRA scores, as assessed in 53 CTRA genes in blood samples [40]. However, other studies have demonstrated associations between an individual’s ACE exposures and their CTRA scores, including a study of 86 women with early-stage breast cancer who had experienced childhood maltreatment [41] and a study of 254 former Nepali child soldiers [42]. Given this larger body of literature, it is not clear why no significant association between maternal ACEs and maternal CTRA was found here. This could be due to the small sample size or the fact that adulthood is a period of less sensitivity to environmental influences than that of childhood [43,44]. 

Consistent with the hypotheses, the ATTACH™ parenting intervention moderated associations between maternal ACEs and CTRA scores, specifically for children, but not as hypothesized. Based on the literature review, it was hypothesized that the ATTACH™ intervention would buffer, flatten, or reduce the association between higher maternal ACEs and higher child CTRA scores. Instead, the interaction was such that an association between maternal ACEs and CTRA scores was only evident in the intervention group, such that low maternal ACEs were associated with lower child CTRA scores. In other words, in contrast with other literature demonstrating the parenting programs that may be most beneficial for those at higher risk [34,35], the ATTACH™ intervention had the most benefit for or impact on child CTRA in children of mothers with low ACEs. Specifically, above maternal ACEs scores of seven, the CTRA scores of children in the intervention group were effectively the same as those of children in the control group. 

Parents with higher ACEs are likely to experience higher adulthood adversity and poorer health and economic circumstances, which may overwhelm their capacity to engage in a parenting intervention or benefit as much as those with lower ACEs [45]. Trauma history has been demonstrated to interfere with the neural processing of reflective function in the prefrontal cortex, so parents with higher ACEs score may be more challenged to practice reflective functioning skills [46]. The ATTACH™ intervention specifically focuses on developing parental reflective functioning; it is possible that mothers with higher ACEs may not experience the same improvements in reflective function or require more time or supports to achieve the same gains as mothers not, or less, affected by ACEs. Future studies should assess whether improved reflective functioning, or other factors (e.g., feeling better supported as a parent, sense of parental efficacy, personal mastery, etc.), could account for the attenuated associations between maternal ACEs and child CTRA scores. These parents may require additional, more intensive, or longer-term trauma-informed approaches. For example, offering supplemental counseling sessions focused on recognizing and responding to past trauma and its impact on parenting for parents with higher levels of ACEs may be useful prior to participation in ATTACH™. With its innovative focus on reflective function, ATTACH™ was designed to supplement parenting and other support programs that families with complex psychosocial needs receive in community agencies [32]. Our finding validates this approach; ATTACH™ may be best delivered in tandem with additional resources, especially for those with more complex trauma and adversity histories, to ensure that parents and children benefit equitably.

In contrast, the ATTACH™ intervention did not moderate associations between maternal ACEs and maternal CTRA scores. It is possible that this is because only a trending or marginal association was observed between maternal ACEs and maternal CTRA scores here, and so there was technically no overall significant association between ACEs and CTRA to moderate. It is also possible that differences in timespans account for the differences. For example, maternal ACEs may have happened too long ago or are too biologically embedded to be undone by a parenting intervention, or benefits could take longer to appear (e.g., months later, rather than immediately after the intervention). It is also possible that the ATTACH™ has different benefits and impacts for mothers as compared to children. Previous research suggests that children benefit from ATTACH™ due to increased maternal reflective functioning and thus more sensitive parenting overall, with broad impacts on child outcomes [32,33]. In contrast, benefits for mothers could be more complex, including increased self-awareness and better relationships with their children, but also through better access to supports, better mental health, and an increased sense of competence, either as a parent or overall. Although ATTACH™ had a beneficial impact on maternal inflammatory gene expression [33], the ATTACH™ mechanisms that benefit maternal health are comparatively less well explored, and it is not clear if they could or are expected to undo the effects of ACE exposures. Additional research is needed to understand how ATTACH™ specifically affects and benefits mothers, both immediately following the intervention and in the longer term.

This study has several limitations. The findings are challenged by the small sample size; while significant associations were observed, the potential for spuriousness remains. Given the small sample size in this pilot clinical trial, we are not able to make strong claims about the associations and interaction effects due to the lack of statistical power and risk of false-positive or -negative claims. Future studies using a prospective, longitudinal design and a larger sample size are recommended to examine the effects of ACEs and interventions on CTRA gene expression among parent–child dyads. Such a study is currently underway [32]. This study focused on a selective sample of mothers and children with exposure to current traumatic experiences of domestic violence who may be more likely to experience ACEs than the general population [47]. This limits the generalizability of the findings to general populations or those with less challenging experiences. It is also important to note that we did not include a comparison group with no exposure to domestic violence. Thus, it is not possible to discern the extent to which children’s CTRA scores can be directly attributable to mothers’ childhood adversities. We recommend future research to replicate these results in other populations with less current traumatic experiences, or include a comparison group to discern the impact of childhood and adulthood adversities on maternal and child CTRA scores. Like many existing studies on ACEs, mothers’ exposure to childhood adversity was retrospectively measured, which also limits the inference on causal relationships between ACEs and CTRA scores. In addition, this study operationalized ACEs exposure to total scores without considering the chronicity and severity of the maternal ACEs exposure. Nevertheless, it is remarkable that both the maternal ACEs and the intervention exert a demonstrable effect on child CTRA in the current study. This study solely focused on the CTRA immune cell gene expression, while childhood adversity challenges, multisystem physiology and parental ACEs may lead to multiple biological (e.g., disruptions in stress hormones, neuro-endocrine–immune–metabolic function, and epigenetics) and behavioral changes potentially impact the health of their children. Future exploratory studies should consider including other biobehavioral processes to further understand the mechanisms of the intergenerational transmission and impact of childhood adversities. Lastly, this study did not examine specific health outcomes relevant to CTRA scores and the ATTACH™ intervention, such that the clinical significance of the results remains to be determined in future research with larger samples. 

## 5. Conclusions

In conclusion, in a high-risk sample of mother–child dyads affected by domestic violence, maternal ACEs were associated with poorer child, but not maternal, CTRA scores. Moreover, the ATTACH™ parenting intervention conferred more protection in relation to children (as evidenced by CTRA scores) when maternal ACEs were low. These findings highlight how parental ACEs could be transmitted among generations and suggest that parenting interventions could moderate intergenerational impact. 

## Figures and Tables

**Figure 1 ijerph-21-00776-f001:**
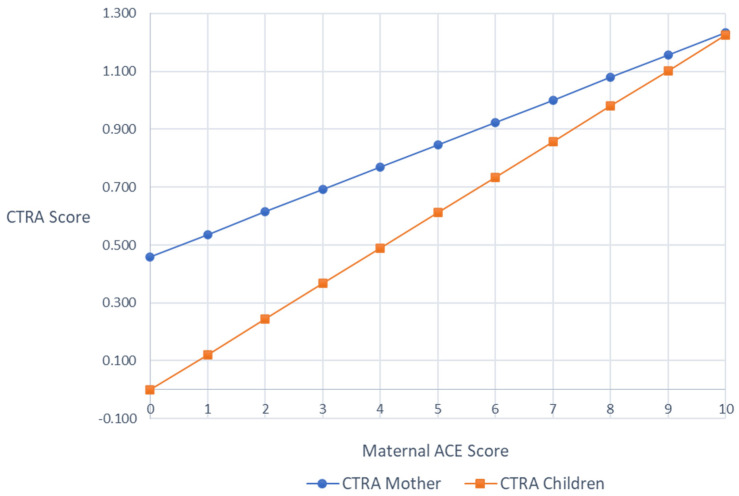
Covariates adjusted mothers’ and children’s CTRA scores by maternal ACEs scores.

**Figure 2 ijerph-21-00776-f002:**
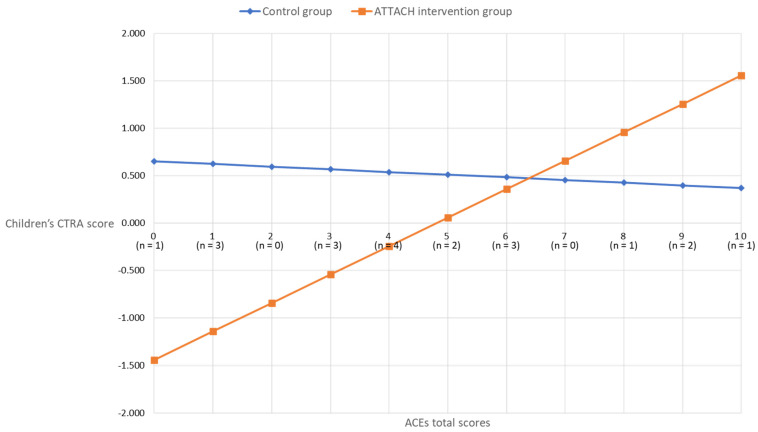
Covariates adjusted children’s CTRA scores by maternal ACEs scores in intervention and control groups.

## Data Availability

Data sharing is not available to this article.

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
