# Peer review of "Childhood Adversities and the ATTACHTM Program’s Influence on Immune Cell Gene Expression"

_ijerph, 2024, doi:10.3390/ijerph21060776_

Round 1

Reviewer 1 Report

Comments and Suggestions for Authors

This is an interesting manuscript addressing a highly relevant topic; however, it presents two main limitations:

  1. The very small sample size, with only 9 dyads in the experimental group and 11 in the control (waiting list). I have doubts about the relevance of findings based on such a limited sample size.

  2. The authors enrolled participants from an anti-violence center, resulting in a highly selected sample with problems related to generalizability. This poin is also particularly significant considering that the authors aim to test whether maternal Adverse Childhood Experiences (ACEs) are associated with mother and/or child CTRA scores, and whether a parenting intervention (ATTACH™) moderates this association. It appears that living in a domestic violence situation is considered solely a maternal adverse experience, whereas it is also and especially adverse for the child. Moreover, a violent romantic relationship can hardly be defined as an "adverse childhood experience," as it is typically faced in adulthood by the mother.

Minor issues:

The methods, participants, etc., should be more extensively reported. It is not sufficient to reference a previously published paper.

Comments on the Quality of English Language

Nothing to highlight

Author Response

1.       This is an interesting manuscript addressing a highly relevant topic; however, it presents two main limitations:The very small sample size, with only 9 dyads in the experimental group and 11 in the control (waiting list). I have doubts about the relevance of findings based on such a limited sample size.

We acknowledged that the findings were challenged by small sample size and the potential for spuriousness in the limitation section. A prospective, longitudinal study with a larger sample size is currently underway and we plan to examine the effects of ACEs and intervention on CTRA gene expression among parent-child dyads in the near future.

2.       The authors enrolled participants from an anti-violence center, resulting in a highly selected sample with problems related to generalizability. This poin is also particularly significant considering that the authors aim to test whether maternal Adverse Childhood Experiences (ACEs) are associated with mother and/or child CTRA scores, and whether a parenting intervention (ATTACH™) moderates this association. It appears that living in a domestic violence situation is considered solely a maternal adverse experience, whereas it is also and especially adverse for the child. Moreover, a violent romantic relationship can hardly be defined as an "adverse childhood experience," as it is typically faced in adulthood by the mother.

To clarify, we measured mothers’ adverse childhood experiences using the original CDC ACE study’s 10-item ACEs scale that captured mothers’ exposure to abuse, neglect, and household challenges in their childhood. We tested whether mothers’ exposure to their childhood adversities are associated with mother and or child CTRA.

Minor issues

The methods, participants, etc., should be more extensively reported. It is not sufficient to reference a previously published paper.

We added more details on participants.

Reviewer 2 Report

Comments and Suggestions for Authors

 Dear Authors,

This manuscript is a very interesting pilot study. It is an interdisciplinary approach of  ACE/immunity/intervention and therefore an innovative perspective. The idea is original, the manuscript is well written but the number of participants is not high and therefore the results not very strong, issue that is addressed in the limitations of the study.

Example.As far as it concerns 3.2. Maternal ACEs and ATTACHTM Intervention Effects on CTRA Gene Expression, Figure 2 and Discussion:  the paragraph is not clear to me and moreover the concluding assumptions seem to me not very strong (Figure 2 and lines 269-276)

2. I would suggest to characterize the sample groups as intervention groups and not as treatment groups because the study is related to an intervention procedure.

Author Response

This manuscript is a very interesting pilot study. It is an interdisciplinary approach of  ACE/immunity/intervention and therefore an innovative perspective. The idea is original, the manuscript is well written but the number of participants is not high and therefore the results not very strong, issue that is addressed in the limitations of the study.

Example. As far as it concerns 3.2. Maternal ACEs and ATTACHTM Intervention Effects on CTRA Gene Expression, Figure 2 and Discussion:  the paragraph is not clear to me and moreover the concluding assumptions seem to me not very strong (Figure 2 and lines 269-276)

We appreciate your positive comments and would like to provide clarifications on Figure 2. Figure 2 shows children’s CTRA scores by mothers’ ACEs scores in the control and intervention group. For the control group (depicted in color blue), the association between children CTRA scores and mothers’ ACEs scores was not significant. For intervention group (depicted in color orange), the association between children CTRA scores and mothers’ ACEs score was significant, such that lower children CTRA scores were associated with lower mothers’ ACEs score, suggesting that the intervention was more effective for children of mothers less affected by early life adversity.   

We are cautious about drawing strong conclusion because of the small sample size which we addressed in the limitation in Discussion.

2. I would suggest to characterize the sample groups as intervention groups and not as treatment groups because the study is related to an intervention procedure.

We changed the wording from “treatment” to “intervention” group throughout the paper.

Round 2

Reviewer 1 Report

Comments and Suggestions for Authors

I appreciate the authors' effort to improve the manuscript by including more information about previous studies and providing additional details on participants and procedures. Nevertheless, the limitations regarding sample size and participant selection are, in my opinion, difficult to overcome. The authors clarified that they asked mothers about childhood traumatic experiences and not current ones. However, I still believe that living in a domestic violence situation is a traumatic experience for both mothers and children, thus this is a potential bias for the study's purpose and its generalizability.

Moreover, as highlighted by the editorial staff, the self-citation rate is over 32%, so the self-citations should be strongly reduced.

Author Response

We appreciate your additional feedback to further improve the quality of this paper and incorporate it into the limitation addressing how our selective sample limits the study generalizability. 

We have reduced self-citation and added new citations.